# Targeting Signalling Pathways in Chronic Wound Healing

**DOI:** 10.3390/ijms25010050

**Published:** 2023-12-19

**Authors:** Lian Bonnici, Sherif Suleiman, Pierre Schembri-Wismayer, Analisse Cassar

**Affiliations:** Department of Anatomy, University of Malta, MSD 2080 Msida, Malta; lian.bonnici.18@um.edu.mt (L.B.); sherif.s.suleiman@um.edu.mt (S.S.); pierre.schembri-wismayer@um.edu.mt (P.S.-W.)

**Keywords:** chronic wound healing, signalling pathways, PI3K/AKT, TGF-β, Notch, HIF-1, Nrf2, Wnt/β-catenin, small molecules drugs, naturally derived compounds, stem-cell-based therapy, oligonucleotide delivery nanoparticles, exosomes, peptide-based platforms

## Abstract

Chronic wounds fail to achieve complete closure and are an economic burden to healthcare systems due to the limited treatment options and constant medical attention. Chronic wounds are characterised by dysregulated signalling pathways. Research has focused on naturally derived compounds, stem-cell-based therapy, small molecule drugs, oligonucleotide delivery nanoparticles, exosomes and peptide-based platforms. The phosphoinositide-3-kinase (PI3K)/protein kinase B (AKT), Wingless-related integration (Wnt)/β-catenin, transforming growth factor-β (TGF-β), nuclear factor erythroid 2–related factor 2 (Nrf2), Notch and hypoxia-inducible factor 1 (HIF-1) signalling pathways have critical roles in wound healing by modulating the inflammatory, proliferative and remodelling phases. Moreover, several regulators of the signalling pathways were demonstrated to be potential treatment targets. In this review, the current research on targeting signalling pathways under chronic wound conditions will be discussed together with implications for future studies.

## 1. Introduction

Chronic wounds are an economic burden to the healthcare system. It is estimated that by 2024, the treatment cost of chronic wounds in the United States will exceed USD 22 billion [1]. The annual expenditure for the treatment of an individual wound ranges from 100 to EUR 10,000 [2]. Currently, such wounds are treated using a combination of debridement (removal of necrotic tissue), dressings, infection control, pain control, grafting, growth factor therapy, negative pressure therapy, ultrasound therapy, hyperbaric oxygen therapy, compression stockings and stem cell therapy [3,4,5].

Although there are several treatment options under investigation, few have been approved by the Food and Drug Administration (FDA) [6]. The phosphoinositide-3-kinase (PI3K)/protein kinase B (AKT), Wingless-related integration (Wnt)/β-catenin, transforming growth factor-β (TGF-β), nuclear factor erythroid 2–related factor 2 (Nrf2), Notch and hypoxia-inducible factor 1 (HIF-1) signalling pathways have been demonstrated to have critical roles during wound healing by modulating cellular proliferation, migration, angiogenesis, inflammation and tissue remodelling [7,8,9,10,11]. However, dysregulated signalling impairs wound healing and promotes chronic wound formation [12]. This review aims to provide a brief overview of the wound healing process, to discuss currently investigated treatments targeting the PI3K/AKT, Wnt/β-catenin, TGF-β, Nrf2, Notch and HIF-1 signalling pathways in wound healing (Figure 1 and Table 1) and to highlight novel targets to mitigate the effect of the dysregulated signalling pathways on chronic wound healing.

### 1.1. The Physiology of Wound Healing

The skin is the largest organ of the human body and is essential for providing a barrier against the external environment. When the skin barrier is disrupted by an insult, wound healing is activated. The process of wound healing can be categorised into four stages: haemostasis, inflammation, proliferation and remodelling (Figure 1). Immediately after a skin injury, haemostasis is initiated to stop bleeding and invasion by pathogens. This process is characterised by vascular contraction, platelet aggregation and fibrin formation. The inflammatory phase is initiated by the sequential migration of neutrophils, macrophages and lymphocytes into the wound owing to platelet-derived growth factor (PDGF), transforming growth factor-beta (TGF-β) and fibroblast growth factor (FGF) signalling [5]. Neutrophils are essential for the clearance of pathogens and cellular debris, whilst macrophages play an essential role in wound healing by secreting several pro-inflammatory cytokines such as Tumour Necrosis Factor-alpha (TNF-α) and interleukin-1 (IL-1) [44]. During the late phase of inflammation, T lymphocytes regulate the inflammatory response by halting the production of interferon-γ and allowing the progression to the proliferative phase [45]. During the proliferative phase, M1 macrophages transform into the M2 subtype which promotes proliferation by secreting anti-inflammatory cytokines and collagen precursors, which in turn stimulate fibroblast proliferation [46]. The role of the proliferative phase is to secure the wound by stimulating the formation of granulation tissue (vascularised tissue that develops after the damping of inflammation), re-epithelialisation and angiogenesis, which supplies blood to the newly formed skin.

Low levels of oxygen in wounds promote angiogenesis by activating hypoxia induced factor-1 (HIF-1) [47]. HIF-1 upregulates vascular endothelial growth factor-A (VEGF-A) expression in endothelial cells [47]. VEGF-A stimulates the growth and branching of existing blood vessels. Macrophages also promote angiogenesis by producing matrix metalloproteinases (MMPs) which degrade the extracellular matrix (ECM)-components and release chemotactic factors such as TNF-α, VEGF and TGF-β [48].

In sync with angiogenesis, re-epithelialisation and granulation tissue formation occur. Keratinocytes play a major role in re-epithelialisation by undergoing epithelial-to-mesenchymal transition (EMT) via signalling by growth factors and cytokines such as FGF, TGF-β, PDGF and epidermal growth factor (EGF) to initiate migration from the wound edge [48]. Then, keratinocytes undergo terminal differentiation to form a thin-layered epithelium. In addition, wound fibroblasts respond to various growth factors and cytokines, including TGF-β and PDGF [48]. This stimulates fibroblasts to form granulation tissue by synthesising several ECM components and differentiating into myofibroblasts, resulting in wound contraction [49].

The final phase of wound healing involves remodelling of the ECM healing [49]. This phase begins two–three weeks after wound formation. However, it can also be initiated for up to one year after wound creation. During remodelling, TGF-β1 signalling stimulates fibroblasts to differentiate into myofibroblasts, which in turn produce collagen types I and III. Collagen fibres are enforced by crosslinking, catalysed by transglutaminases and lysyl oxidases. At a later stage of remodelling, MMPs degrade type III collagen resulting in scar formation with an abundance of type I collagen. The resulting healed wounds have a lower tensile strength than unwounded skin and the maximum tensile strength achieved by the wounded skin is 80% that of unwounded skin.

### 1.2. Pathophysiology of Wound Healing

Abnormal wound healing can result in chronic wounds which fail to achieve complete closure. Some common features of chronic wounds include exacerbated inflammation (Figure 2), recurrent infections, poor blood supply, pain and development of necrotic tissue [4]. Furthermore, chronic wounds can be classified into four primary categories: diabetic foot ulcers (DFUs) typically found in individuals with diabetes, as well as venous ulcers, pressure ulcers, and arterial insufficiency ulcers, which are commonly seen in older individuals and those with limited mobility, such as bedridden patients [1].

Wound healing in chronic wounds may be blocked in the inflammatory and proliferation phases. Infected chronic wounds are blocked in the inflammatory phase and are characterised by excessive neutrophil infiltration which produces ROS and serine proteases, which damage ECM components and degrade essential growth factors such as TGF-β and PDGF [50]. Blockage in the proliferation phase may occur due to the failure of keratinocytes, fibroblasts and endothelial cells to respond to growth factors and cytokines, leading to a reduction in cellular proliferation [51]. Therefore, blockage in the proliferation phase leads to failure of wound closure and poor blood supply due to impairment of angiogenesis.

## 2. The PI3K/AKT Pathway

### 2.1. The PI3K/AKT Signalling Pathway in Wound Healing

The PI3K/AKT pathway is initiated by the binding of cytokines and growth factors such as PDGF and epidermal growth factor (EGF) to receptor tyrosine kinases (RTKs) [52]. RTKs undergo autophosphorylation of tyrosine residues in the cytosolic domain. PI3K then binds to the activated RTKs via Src homology 2 domains. The activated PI3K phosphorylates phosphatidylinositol 4,5-bisphosphate (PIP2) to phosphatidylinositol-3,4,5-triphosphate (PIP3). AKT is activated through phosphorylation by 3-phosphoinositide-dependent kinase 1 (PDK1) and mammalian target of rapamycin C2 (mTORC2), followed by binding with PIP3. Phosphorylated AKT leads to the activation of mTORC1, which in turn promotes the translation of several pro-migratory and proliferative RNA transcripts. In normal wound healing, PI3K/AKT signalling enhances EMT, cell proliferation and angiogenesis and reduces inflammation [16,53,54].

PI3K/AKT signalling peaks during the inflammatory and proliferative phases of wound healing [55]. Its inhibition results in impaired wound healing [11]. Several studies have explored treatments that promote wound healing by activating the PI3K/AKT pathway.

### 2.2. Natural Extracts Promoting the PI3K/AKT Pathway

Natural extracts have active components that promote wound healing by activating PI3K/AKT signalling. Ruttanapattanakul et al. reported that the active component of *Boesenbergia rotunda* extract, kaempferol, enhanced the proliferation and migration of HaCaT cells via PI3K/AKT activation as demonstrated by an increase in the level of phosphorylated AKT [56]. Xiao et al. reported that ozone oil (oil produced by bubbling ozone into oil, such as olive oil) promoted wound healing in mice by enhancing the migration of fibroblasts via PI3K/AKT signalling as showed by an increase in the levels of phosphorylated mTOR and AKT [16]. Furthermore, ozone oil reduced inflammation by upregulating the anti-inflammatory interleukin-6 (IL-6) and downregulating the pro-inflammatory cytokine TNF-α. Fucoidan, a sulphated polysaccharide extracted from brown algae has been tested on human umbilical vein endothelial cells (HUVECs) via in vitro wound healing assays, resulting in increased migration by inducing the phosphorylation of AKT [24]. Similarly, naringin, the active component of *Drynaria fortune*, induced the proliferation of progenitor endothelial cells isolated from rats and promoted angiogenesis through phosphorylation of AKT [25]. In a study by Huang et al., ginsenoside Rg1, an active component of *Panax ginseng*, promoted the healing of DFUs in diabetic Sprague–Dawley rats by activating PI3K/AKT signalling as reported by an increase in the levels of phosphorylated PI3K and AKT [57]. Even though these studies show the importance of extracts in wound healing, cell lines and rodent models have their limitations. In the case of rodent models, wound contraction results in rapid healing when compared to humans. Also, transcriptome and protein analysis could be carried out to confirm the activation of the PI3K/AKT pathway.

### 2.3. Stem-Cell-Based Treatment Promoting the PI3K/AKT Pathway

Stem cells have a crucial role during wound healing as they exhibit multi-lineage differentiation and secrete several growth factors such as VEGF, EGF and keratinocyte growth factor, which promote wound healing progression [58]. Recently, stem cell therapies have also been investigated for their effects on chronic wound healing. Exosomes derived from hypoxic adipose stem cells (ASCs) have been shown to contain several miRNAs such as miR-21-3p that promote in vitro proliferation and migration of fibroblasts by activating the PI3K/AKT signalling pathway, marked by an increase in the protein level of phosphorylated AKT [17]. Moreover, treatment with exosomes derived from hypoxic ASCs promoted wound healing in diabetic mice [17]. Similarly, Xiu et al. demonstrated that exosomes derived from human umbilical vein mesenchymal stem cells (HUC-MSCs) loaded with microRNA (miRNA)-150-5p promote the proliferation and migration of hydrogen peroxide-injured HaCaT cells via PI3K/AKT activation [59]. Exosomes produced from human amniotic epithelial stem cells contain multiple miRNAs, which increase HUVEC and fibroblast proliferation and wound healing in diabetic rats by activating the PI3K/AKT pathway as shown by increased levels of phosphorylated AKT [15]. However, more research is required to determine the effect of miRNAs on the expression of proteins in the PI3K/AKT pathway. Furthermore, since the expression of miRNAs is influenced by epigenetics, using epigenetic modifying agents such as DNA demethylating agents and histone deacetylase inhibitors could be employed to treat stem cells to induce the expression of therapeutic miRNAs.

## 3. The Canonical Wnt/β-Catenin Pathway

### 3.1. Overview of the Wnt/β-Catenin Pathway

The canonical Wnt/β-catenin signalling pathway is activated by the binding of Wnt ligands to frizzled transmembrane receptors [9]. Frizzled receptors, together with the low-density lipoprotein receptor-related protein 5/6 co-receptor complex (LRP5/6), recruit downstream proteins, including dishevelled (Dvl) and axin. As a result, the destruction complex composed of axin, adenomatous polyposis coli (APC), glycogen synthase kinase-3 (GSK-3) and casein kinase 1 (CK1) is segregated. Then, β-catenin is released from the destruction complex and translocates to the nucleus, where it binds to T-cell factors/lymphoid transcription factors (TCFs/LEFs). In the absence of Wnt ligand binding, β-catenin is phosphorylated by CK1 and GSK-3. Phosphorylated β-catenin recruits E3-ubiquitin ligases, targeting it for proteasomal degradation.

During wound healing, the Wnt/β-catenin pathway is activated by hypoxia as it drives the expression of several Wnt ligands [60]. The Wnt/β-catenin signalling pathway enhances wound healing by upregulating the expression of genes involved in cellular proliferation, migration, angiogenesis and stem cell activation [61,62,63]. The inhibition of the Wnt/β-catenin pathway in diabetic mice caused by the overexpression of pigment epithelium-derived factor (PEDF) resulted in wound healing impairment by inhibiting angiogenesis [64].

### 3.2. Targeting Wnt/β-Catenin Signalling to Enhance Chronic Wound Healing

Numerous approaches for improving wound healing, such as naturally sourced medications, stem-cell-based treatments, tetrahedral framework nucleic acids and shock wave therapy, were identified as being effective in stimulating the Wnt/β-catenin pathway. Wu et al. demonstrated that ruyi jinhuang powder accelerated wound healing in diabetic-induced mice by promoting angiogenesis, proliferation and migration of dermal fibroblasts via the activation of the Wnt/β-catenin pathway as demonstrated by the increase in the protein levels of β-catenin [26]. Nie et al. utilised a combinatorial treatment approach by constructing a gel composed of asiaticoside with nitric oxide [18]. This gel was reported to promote the in vitro proliferation and migration of human foreskin fibroblasts and enhance the healing of diabetic cutaneous ulcers in mice via an increase in the gene expression of *Wnt1*, *Wnt4* and *β-catenin*. The limitations in mice or rodent models could possibly be addressed by using other animal models such as rabbits. Rabbits undergo wound healing by re-epithelialisation which is similar to wound healing in humans. Additionally, naturally sourced medications can be incorporated into drug delivery systems such as lipospheres and nanoparticles to increase their effectiveness.

Stem cell therapy has been shown to promote chronic wound healing by increasing Wnt/β-catenin signalling. The acellular conditioned medium (CM) of ASCs was investigated as a stem-cell-based approach to wound healing. Guo et al. investigated the application of conditioned medium of ASCs on human ex vivo wound models [65]. The researchers reported that the conditioned medium of ASCs promoted wound healing by upregulating the expression of PDGF-AA and hepatocyte growth factor (HGF). Moreover, the Wnt/β-catenin pathway was activated as demonstrated by an increase in the expression of β-catenin. Although this study showed that the CM of ASCs can promote wound healing, the active components present in the CM were not characterised. Hence further investigations are warranted to identify its active components.

Tetrahedral framework nucleic acids (TFNAs) have been applied as a novel treatment strategy to enhance wound healing either by acting as a vehicle or by direct application to the wound. TFNAs are composed of four single-stranded DNAs assembled into a three-dimensional structure [66]. TFNAs can enter cells via caveolin-mediated endocytosis and escape lysosomal degradation [66]. The in vivo application of TFNAs in chronic diabetic wounds promoted healing by upregulating VEGF-A and thus boosting angiogenesis [27]. Furthermore, treatment with TFNAs promoted re-epithelialisation via Wnt/β-catenin signalling as showed by an increase in the expression of β-catenin, LEF1 and TCF1 [27]. In addition to diabetic wounds, TFNAs should also be investigated for the effect on the healing of other types of chronic wounds including pressure, venous and arterial insufficiency ulcers. Different single-stranded DNA sequences should be examined since unique combinations may have a larger effect than previously created TFNAs.

Currently, extracorporeal shock wave therapy (ESWT) is applied to treat conditions including plantar fasciitis, Achilles tendinopathy and calcifying tendinitis [67]. ESWT involves the use of sound waves to promote wound healing. Studies show that ESWT may improve chronic wound healing and reduce scar formation [68,69]. A study by Chen et al. demonstrated that ESWT promoted healing of chronic wounds in diabetic-induced mice by upregulating the expressions of β-catenin, Wnt1, Wnt3a, Wnt4 and Wnt5a [70]. Hence confirming that ESWT may promote chronic wound healing by promoting Wnt/β-catenin signalling. Since ESWT is not a small-molecule-based treatment, in a clinical scenario, it would be ideal to investigate the combination of ESWT with small-molecule-based treatments to invoke an additive or synergistic effect.

## 4. TGF-β Signalling

### 4.1. Overview of the TGF-β Signalling Pathway

The TGF-β family consists of TGF-β1-3, bone morphogenic proteins and activins [71]. TGF-β signalling is initiated by the binding of TGF-β to TGF-β receptor 2 (TβRII). TβRII, then forms a heterotetrameric complex consisting of two pairs of TβRII and one pair of TGF-β receptor 1 (TβRI) in which TβRI is activated by phosphorylation of serine/threonine residues in the glycine/serine domain. Next, the activated TβRI activates the suppressor of mothers against decapentaplegic (Smad) 2 and Smad3 by phosphorylation. Finally, the Smad2/3 interacts with Smad4, and the Smad complex translocates to the nucleus where it modulates the transcription of genes such as integrins, E-cadherin, c-Myc and IL-6 [72]. TGF-β is secreted as an inactive form with latency-associated peptide prodomain (LAP) bound to it. For TGF-β signalling to take place, LAP is cleaved by MMPs and by conformational changes induced by interactions with integrins [73].

TGF-β has a critical role in the wound healing process. In the inflammatory phase, TGF-β secreted by platelets and leukocytes acts as a chemokine by promoting the proliferation and migration of fibroblasts to the wound site [43,74]. Moreover, during the proliferative phase, TGF-β1 promotes the migration of keratinocytes and enhances angiogenesis and granulation tissue formation [43]. During the remodelling phase, TGF-β1 stimulates the differentiation of fibroblasts to myofibroblasts as shown by the upregulation of α-smooth actin, hence promoting wound contraction and production of collagen [75].

### 4.2. Targeting TGF-β Signalling to Enhance Chronic Wound Healing

Excessive TGF-β1-2 expression in myofibroblasts causes hypertrophic scar formation by inducing the over-expression of collagen [76]. Therefore, reducing the expression of TGF-β1 diminishes hypertrophic scar formation. The application of *Spirulina platensis* on burns and full thickness wounds of rat models resulted in less scar formation by down-regulating TGF-β1 during the remodelling phase [36]. Furthermore, in mice burn skin models, treatment with exosomes derived from microRNA-29a (miR-29a)-modified adipose-derived MSCs caused the downregulation of TGF-β2, which in turn reduced scar formation by inhibiting the TGF-β2/Smad3 signalling [77]. In contrast, upregulation of TGF-β3 has been shown to accelerate wound healing of vocal fold mucosal and skin injuries whilst also reducing scar formation [37,78]. 

Recently, nesprin-2 and histone deacetylase 5 (HDAC5) have been identified as potential novel targets for reducing hypertrophic scarring by attenuating TGF-β1 signalling. Silencing of nesprin-2 reduced the expression of TGF-β1 and delayed the differentiation of fibroblasts into myofibroblasts, as shown by the downregulation of α-SMA and collagen type I [79]. HDAC5 is overexpressed in hypertrophic scars [80]. In mouse models, HDAC5 knockout using the CRISPR/Cas9 system reduced scar formation by inhibiting Smad2 and Smad3 phosphorylation, thus attenuating TGF-β signalling [80].

In contrast to hypertrophic scars, TGF-β1 expression is reported to be downregulated in chronic wounds [73]. Therefore, research focused on increasing TGF-β1 expression to promote chronic wound healing should be further explored. Several treatments have been reported to increase the expression of TGF-β1. Therapies such as maggot therapy, naturally derived treatments, stem-cell-derived treatments, exposure to electromagnetic fields and oxygen therapy have been examined. Maggot therapy is used as a method of debridement (removal of necrotic tissue) to expose healthy tissue and promote wound closure [81]. Maggot therapy was demonstrated to increase the expression of TGF-β1 in DFU tissue isolated from patients with type 2 *diabetes mellitus* [82]. Furthermore, granulation tissue formation was enhanced [82]. To confirm the mechanism of action of maggot therapy, it would be ideal to perform RNA sequencing and mass spectrometry on treated DFUs to determine how maggot therapy influences TGF- β signalling. 

Another treatment utilises the Chinese herbal medicine, Huiyang shengji. It is composed of cinnamomi cortex, ginseng radix et rhizome, Chuanxiong rhizoma and cervi cornu pantotrichum and it has been shown to promote chronic wound healing of DFUs in mice by enhancing TGF-β signalling [83]. Treatment of human skin fibroblasts (HSF) stimulated with CM from M1 macrophages with Huiyang Shengji ameliorated the effect of the pro-inflammatory state induced by the CM of M1 macrophages by enhancing the expression of pSmad3, pC-Jun and C-Fos, therefore reducing inflammation [84].

Similarly, the extract from indigo leaves was also shown to improve healing of full-thickness wounds in rats by promoting angiogenesis and re-epithelialisation, while also diminishing inflammation [19]. These effects were attributed to an increase in TGF-β1 expression which regulates the proliferative and inflammatory phases of wound healing [19]. Since the extract of indigo leaves was only tested on a rat excisional model of acute wound healing, its effect on chronic wounds was not investigated. Therefore, human skin equivalents may be used to test the effect of the indigo leaf extract under chronic wound conditions.

Similarly, it was demonstrated that treatment of wounds of streptozotocin-induced diabetic mice with caffeic acid phenethyl ester (CAPE) nanoparticles promoted wound healing by enhancing wound contraction, collagen formation and TGF-β1 expression while also decreasing the expression of the pro-inflammatory markers, IL-6 and TNF-α [20].

Pirfenidone is used as an anti-fibrotic drug to reduce the expression of pro-fibrotic genes including TNF-α, type I collagen, PDGF and TGF-β1 [38]. In a randomised double-blinded clinical trial, a combination of pirfenidone and the antimicrobial diallyl disulfide oxide was shown to promote healing of DFUs and increase the expression of TGF-β1 and TGF-β3 after two months of treatment [85].

Other naturally derived treatments which promote wound healing by enhancing the expression of TGF-β1 have also been described. Examples of treatments include β-acetoxyisovaleryl alkannin, polyherbal formulations and the flavonoids, vicenin-2 and hesperidin [39,40,41,42]. Polyherbal formulations are composed of several Thai plant extracts including *Curcuma longa* Linn and *Zingiber cassumunar* Roxb. Treatment of HaCaT with the polyherbal formulations under high-glucose conditions promoted the migration of HaCaT cells characterised by the increase in the expression of TGF-β1 [39]. Further in vivo studies are required to confirm the effect of such polyherbal formulations on chronic wound healing. Treatment of DFUs in Sprague–Dawley mice with hesperidin (100 mg/kg) and vicenin-2 (50 µM) promoted the proliferation of fibroblasts, re-epithelialisation and angiogenesis [40,41]. Similarly, treatment of pressure-induced venous ulcers of rabbits with β-acetoxyisovaleryl alkannin increased the rate of wound closure. The positive effects of these treatments were attributed to an increase in the expression of TGF-β1 [39,40,41].

Continuous diffusion of oxygen has been investigated as a novel treatment strategy to enhance wound healing and TGF-β expression. DFUs treated with continuous diffusion of oxygen showed improved wound closure and an increase in TGF-β expression, albeit not significant [86]. Some of the limitations in certain studies are the small number of participants, the levels of growth factors including TGF-β1 were not measured until complete wound closure and not all controls were included.

Recently, the WD Repeat-Containing Protein 74 (WDR74) was shown to promote mouse M2 macrophage polarisation by activating the TGF-β/Smad signalling pathway [13]. Therefore, the development of a treatment that enhances WDR74 expression would provide a novel approach to promote wound healing by allowing progression from the consistent inflammatory state present in diabetic wounds to the proliferative phase.

## 5. Nrf2 Signalling Pathway

### 5.1. Overview of the Nrf2 Signalling Pathway

Nrf2 is a member of the transcription factor family Cap’n’Collar which also includes NFE2, Nrf1, Nrf3, Bach1 and Bach2 [87]. Nrf2 protects cells from oxidative damage induced by ROS by promoting the expression of antioxidant proteins and detoxification enzymes [88]. During cellular homeostasis, the major regulator of Nrf2, Kelch-like ECH-associated protein 1 (Keap1), binds to Nrf2 and targets it for proteasomal degradation [89]. However, at high ROS levels, the thiol groups of the cysteine residues of Keap1 are oxidised. Consequently, Keap1 undergoes a conformational change, which renders its binding with Nrf2 unfeasible [89]. Free Nrf2 translocates to the nucleus, forms a heterodimer with MAF proteins and then the Nrf2-MAF complex binds to the antioxidant response elements to enhance the transcription of Nrf2 target genes [89]. The gene targets of Nrf2 include glutathione synthase, superoxide dismutase 1 and catalase [89].

The role of Nrf2 during wound healing is to promote epithelial proliferation and migration and reduce oxidative stress and apoptosis [90]. Since chronic wounds are characterised by high ROS levels, targeting the Nrf2 signalling pathway is a satisfactory approach for reducing oxidative stress and promoting chronic wound healing [91,92].

### 5.2. Targeting Nrf2 Signalling to Enhance Chronic Wound Healing

The human placental extract (HPE) activates the Nrf2 signalling pathway. HPE contains uracil, tyrosine, phenylalanine, tryptophan and collagen-derived peptides [93]. Although HPE was not investigated on chronic wound healing, HPE was demonstrated to reduce cell senescence and oxidative stress in fibroblasts treated with H_2_O_2_ by promoting Nrf2 protein expression, which in turn promoted the expression of several antioxidant genes including *APOE* and *PTGS1* [94]. Therefore, this study demonstrated that HPE may have the potential to promote chronic wound healing by reducing oxidative stress. Furthermore, the identification of the active constituents of HPE would be ideal to further investigate the mechanism of action and would also help in the mass production of the active constituents.

Recently, olive oil has been investigated on skin wounds for its anti-inflammatory effects [95]. Olive oil was found to promote the survival and migration of mouse dermal fibroblasts by stimulating Nrf2 activity [96].

Other naturally derived treatments were also shown to promote chronic wound healing by activating Nrf2 signalling. These treatments include the flavonoids: xanthohumol, procyanidin B2, pterostilbene, curcumin, resveratrol and gallocatechin in addition to other naturally derived treatments: dimethyl fumarate, cordyceptin, bee venom, puffball spores, sodium danshensu and paeoniflorin [21,40,97,98,99,100,101,102,103,104,105,106]. There are a few drawbacks that are limiting the use of naturally derived treatments in the clinic. Firstly, more research is needed to identify the active components of olive oil, bee venom and puffball spores that promote wound healing via Nrf2 signalling. Secondly, not all naturally derived treatments were tested on chronic wound conditions and hence their effectiveness has not yet been confirmed. It would also be worthy to investigate the effect of such treatments on the expression of the negative regulators of Nrf2, GSK3-β and Keap1 as this would provide an alternative approach to target the Nrf2 signalling pathway.

RTA 408 is a semi-synthetic oleanane triterpenoid possessing antioxidant properties. Rabbani et al. demonstrated that wounds of type II diabetic mice treated with 0.1% RTA 408 achieved faster wound closure, promoted the formation of granulation tissue and reduced oxidative stress by inducing the expression of Nrf2, manganese superoxide dismutase (SOD), haem oxidase 1, glutathione S-transferase and glutamate cysteine-ligase [107]. This study showed that RTA 408 is a potent antioxidant. Therefore, the combination of RTA 408 with other anti-inflammatory drugs may provide a greater anti-inflammatory effect in chronic wounds and promote the proliferative phase.

Exosomes derived from different stem cell types have been shown to have positive effects on wound healing by activating the Nrf2 signalling pathway. Li et al. demonstrated that exosomes derived from ASCs overexpressing Nrf2, reduced ROS levels and the expression of several pro-inflammatory cytokines in endothelial progenitor cells [108]. Furthermore, the application of exosomes derived from ASCs with normal and overexpressed Nrf2 combined with endothelial progenitor cells in the wounds of streptozotocin-induced diabetic mice was shown to improve the rate of wound closure. Thus, confirming that the overexpression of Nrf2 is essential to promote diabetic wound healing.

Similarly, exosomes overexpressing circ-itchy E3 ubiquitin protein ligase (ITCH) and derived from bone marrow stromal cells promoted wound healing in streptozotocin-induced diabetic mice and alleviated ferroptosis by inducing the expression of Nrf2 [109]. Additionally, ITCH and TAF15 were found to stabilize the mRNA of Nrf2. Hence, further investigations to identify the relationship between ITCH and TATA-binding protein-associated factor 2N (TAF15) are required as they may be novel therapeutic targets to promote chronic wound healing. Finally, exosomes derived from umbilical-cord-derived MSCs with overexpressed non-coding RNA, circ-homeodomain interacting protein kinase 3 (HIPK3) promoted angiogenesis via the inhibition of miR-20b-5p and the activation of Nrf2 in streptozotocin-induced diabetic mice [110].

Ubiquitin-like with PHD and ring finger domains 1 (UHRF1), miR-181b-5p and Keap1 have been investigated as potential targets for promoting Nrf2 signalling. UHRF1 is an epigenetic modifier that modulates DNA methylation [111]. UHRF1 is a positive regulator of Nrf [111]. However, the exact mechanism by which UHRF1 promotes Nrf2 activation remains unknown. Wang et al. demonstrated that exosomes derived from DFUs and overexpressing miR-181b-5p downregulated the expression of Nrf2 [112]. Furthermore, inhibition of miR-181b-5p in DFUs of streptozotocin-induced diabetic mice improved wound healing when compared with the control group (diabetic mice treated with exosomes without the miR-181b-5p inhibitor). Therefore, the downregulation of miR-181b-5p in chronic wounds may enhance the rate of wound closure.

It would be beneficial to carry out sequencing of miRNAs together with computational methods to identify their targets and effects on signalling pathways. Targeting the regulators of Nrf2 expression and activation may also be another approach to promote chronic wound healing. For example, the knockdown of Keap1 with siRNAs packaged into exosomes accelerated wound healing in streptozotocin-induced diabetic mice by increasing the expression of the Nrf2 target, haem oxygenase 1, which is known to reduce inflammation [113].

## 6. Notch Signalling Pathway

### 6.1. Overview of Notch Signalling Pathway

The Notch signalling pathway regulates stem cell differentiation, apoptosis, proliferation and inflammation [114]. In mammals, there are five Notch ligands, four of which are activators and include Delta-like ligand (DLL)-1, DLL-4, Jag1 and Jag2 [115]. The inhibitor of Notch signalling is DLL-3. Furthermore, there are four Notch receptors, Notch1, Notch2, Notch3, and Notch4. Notch receptors are single-pass transmembrane proteins containing an extracellular epidermal growth factor-like domain [115]. The Notch signalling pathway is activated by the binding of Notch receptors to Notch ligand activators expressed by adjacent cells. Next, the Notch receptor is enzymatically cleaved twice by a Disintegrin and metalloproteinase domain 10 (ADAM10) and γ-secretase. As a result, the Notch intracellular domain (NICD) is released into the nucleus where it interacts with the transcription factor centromere-binding factor 1 (CBF1), CBF1 Suppressor of Hairless Lag-1 (CSL) and other co-factors to induce the expression of basic helix-loop-helix (bHLH) transcription factors of hairy/enhancer of split (Hes) family and hairy/enhancer of split related with YRPW motif (Hey) family [116].

### 6.2. Role of Notch Signalling during Normal Wound Healing

During wound healing, the Notch signalling pathway modulates several cell types and processes including inflammation, angiogenesis, epidermal stem cell differentiation and scar formation. During the early stages of normal wound healing, low levels of Notch signalling allow the expression of IL-36α, which is essential to enhance inflammation [117]. At the late stage of wound healing, activation of Notch signalling is essential for promoting M2 macrophage recruitment. Qin et al. showed that applying the recombinant protein collagen triple helix repeat containing 1 (CTHRC1) to wounds in mice undergoing acute healing facilitated the recruitment of M2 macrophages on the seventh day after wound initiation [118]. This effect was achieved by triggering the expression of Notch downstream targets, specifically Hes and Hey. Notably, the positive outcomes of CTHRC1 treatment were nullified when DAPT, an inhibitor of Notch signalling, was administered. This underscores the crucial role of Notch signalling in recruiting M2 macrophages during wound healing.

In keratinocytes and endothelial cells, the Notch signalling pathway is essential for promoting proliferation. Treatment of human neonatal immortalized keratinocytes (NIKS) and primary keratinocyte GS-1-Ep cells with DAPT resulted in a reduction in the proliferation of both cell types [119]. Furthermore, in HUVECs, activation of Notch signalling by Jag1 promoted their migration, proliferation and formation into blood vessels. Thus, this showed that the activation of Notch signalling also promotes angiogenesis [8]. Notch1 signalling also regulates the proliferation and differentiation of epidermal stem cells (EpSCs). The activation of Notch signalling in EpSCs isolated from the skin of rodents by treatment with Jag1 caused an increase in cellular proliferation and inhibited their differentiation to keratinocytes and myofibroblasts [120].

Studies have shown that Notch signalling during wound healing reduces scar formation. The treatment of rabbit ear wounds with FGF promoted wound closure and activated the Notch signalling pathway by upregulating the expression of Notch1 and Jag1 [121]. Activated Notch signalling inhibited scar formation by blocking the differentiation of EpSCs into myofibroblasts [121]. Moreover, Patel et al. showed that the loss of Notch signalling in RNA binding protein-J (RBP-J)^fl/fl^/Cdh5-cre^ERt2^ROSA-YFP mice promoted the mesenchymal transition of endothelial cells to myofibroblasts, which in turn led to fibrosis and reduced wound healing [122].

### 6.3. Dysregulated Notch Signalling in Diabetic Wounds and Hypertrophic Scars

Diabetic wounds and hypertrophic scars are characterised by upregulation in the Notch signalling pathway [123,124]. Narayanan et al. demonstrated that high glucose levels resulted in excessive Notch signalling both in vivo and in vitro and impaired wound healing in streptozotocin-induced diabetic mice [125]. Furthermore, excessive Notch signalling in macrophages has also been identified in type II diabetic mouse models, and inhibition of Notch signalling in macrophages activated by lipopolysaccharide resulted in reduced expression of the pro-inflammatory markers IL-1β and TNF-α [126]. Furthermore, knockout of Notch1 in diabetic KRT14–Cre;Notch1^fl/fl^ mice enhanced angiogenesis and granulation tissue formation in diabetic wounds. Therefore, it can be inferred that excessive Notch signalling in macrophages may cause the hyperinflammatory state present in chronic diabetic wounds.

Shao et al. reported that excessive Notch signalling was also present in fibroblasts isolated from DFUs, which showed higher expression of Notch1-4, Jag1-2, Dll-1, Dll-3, Dll-4, Hes-1 and Hey-1 [124]. Furthermore, the gain-of-function of Notch1 in mice impaired wound healing by inhibiting the differentiation of fibroblasts to myofibroblasts, angiogenesis and IL-6 expression.

Upregulated Notch signalling also causes pathological scarring. Li et al. demonstrated that, excessive Notch signalling in keratinocytes enhanced the expression of several pro-fibrotic genes including TGF-β1, TGF-β2, insulin-like growth factor 1 (IGF-1), EGF, VEGF and connective tissue growth factor (CTGF) and induced abnormal differentiation [123]. Furthermore, a skin incision in mice with RBP-J knockout showed a reduction in Notch signalling and scarring compared to the control [89]. These studies show that targeting excessive Notch signalling in keratinocytes and macrophages may be an ideal treatment strategy to reduce hypertrophic scarring.

### 6.4. Targeting Notch Signalling to Enhance Healing of Chronic Wounds

#### 6.4.1. Inhibition of Notch Signalling to Promote Diabetic Wound Healing

Stem-cell-derived treatments, peptide-based treatments and small molecules are being investigated for their potential to enhance chronic wound healing by targeting the Notch signalling pathway. Inhibition of Notch signalling promotes the healing of diabetic wounds. Treatment of wounds in streptozotocin-induced diabetic mice with a combination of adipose MSCs and platelet-rich plasma accelerated wound healing by promoting angiogenesis and reducing inflammation, scar formation and the expression of Notch1 [28]. According to our current knowledge, small molecule Notch inhibitors are not being investigated as a treatment for chronic wounds. Thus, development of small molecule Notch inhibitors is important to provide an additional approach to promote chronic wound healing.

#### 6.4.2. The Activation of Notch Signalling to Enhance the Healing of Pressure Ulcers and Chronic Limb Ischaemia Ulcers

In contrast to diabetic wounds, activation of Notch signalling promotes the healing of pressure ulcers. Treatment of pressure ulcers in mice with tazarotene nanoparticles loaded in poly lactic-co-glycolic acid, resulted in increased angiogenesis and activation of Notch signalling as shown by an increase in the expression of Notch1, DLL-4, vascular endothelial growth factor receptor (VEGFR) 2 and VEGFR3 [127]. Similarly, activation of Notch signalling promotes the healing of chronic limb ischaemia (CLI) ulcers. Treatment of CLI ulcers in mice with thymosin-β4 promoted wound closure by increasing angiogenesis via activation of Notch signalling [128]. Collectively, these studies show that the activation of Notch signalling in pressure and CLI ulcers can promote wound closure, whereas inhibition of Notch signalling in chronic diabetic wounds may be required for complete wound healing.

#### 6.4.3. Novel Regulators of Notch Signalling as Potential Targets to Enhance Chronic Wound Healing

Recently, miR-200b, miR-203 and kallikrein-binding protein (KBP) were found to regulate Notch signalling under diabetic conditions and may be potential drug targets to enhance diabetic wound healing. Inhibition of miR-200b in HUVEC cells cultured in high glucose concentrations upregulated the expression of Notch1, Jag1 and Hes1 [129]. In addition, the inhibition of miR-200b enhanced the migration of HUVECs and blood vessel formation [129]. Thus, this showed that inhibition of miR-200b may promote diabetic wound healing by increasing angiogenesis via the activation of Notch signalling. Similarly, inhibition of miR-203 in streptozotocin-induced diabetic mice promoted the healing of diabetic wounds by inducing the expression of Notch signalling targets [127]. As a result, Notch signalling activity restored the proliferation of ESCs. KBP is a serine proteinase inhibitor, and its function is to promote vasodilation and inhibit angiogenesis and antioxidative stress [130]. KBP was reported to be highly expressed in patients with DFUs and microvascular complications [131]. Feng et al. demonstrated that from day 7 post-wounding, in streptozotocin-induced diabetic mice, KBP activated the Notch signalling pathway which consequently promoted the polarisation of M1 macrophages [14]. However, inhibition of KBP in RAW264.7 cells with KBP antibody inhibited Notch signalling and blocked the polarisation of M1 macrophages. Considering that Notch signalling has an essential role during wound healing and is dysregulated in chronic wounds, the Notch signalling pathway is a potential drug target to enhance chronic wound healing. In addition, further research is required to identify novel regulators of the Notch signalling pathway under chronic wound conditions to provide further opportunities for treatment development.

## 7. HIF-1 Signalling Pathway

### 7.1. Overview of HIF-1 Signalling Pathway

Under hypoxic conditions, cells activate several signalling pathways to survive. One of the signalling pathways that are activated during hypoxia is the HIF-1 signalling pathway. HIF-1 is a basic helix-loop-helix transcription factor and a dimer of HIF-1α and HIF-1β subunits [132]. During normoxic conditions, proline residues 402 and 564 of the HIF-1α subunit are hydroxylated by the hypoxia-inducible factor prolyl hydroxylase domain enzymes [133]. As a result, the von Hippel–Lindau (VHL), which makes part of an E3 ubiquitin ligase complex, is recruited [133]. The E3 ubiquitin ligase together with E1 and E2 complexes cause the polyubiquitination of HIF-1, hence targeting it for proteasomal degradation [134]. To further reduce the activity of HIF-1α, asparagine 803 is also hydroxylated by factor-inhibiting HIF-1 (FIH-1) [135]. The hydroxylation of asparagine 803 inhibits the interaction of HIF-1α with p300/cyclic adenosine monophosphate response element binding protein binding protein (CBP) [134]. In hypoxic conditions, HIF-1α is not hydroxylated [136]. As a result, HIF-1α is translocated to the nucleus where it dimerises with HIF-1β [137]. The HIF-1 dimer interacts with p300/CBP and then HIF-1 binds to the hypoxia-responsive elements (HREs) of the promoters or enhancers of target genes [137]. The target genes of HIF-1 include VEGF, erythropoietin (EPO), stromal cell-derived factor 1 (SDF-1), glucose transporters 1/3 (GLUT1/3), lactate dehydrogenase-A (LDH-A) and haem oxygenase 1 (HO-1) [138].

### 7.2. Role of HIF-1 Signalling during Wound Healing

During wound healing, HIF-1 signalling is activated as a response to tissue hypoxia where its role is to modulate angiogenesis and inflammation. HIF-1 signalling induces angiogenesis by enhancing the expression of pro-angiogenic genes including VEGF, angiopoietin 2, SDF-1, nitric oxide synthases and adrenomedullin [133]. During wound healing, VEGF is produced by several cell types including keratinocytes, fibroblasts, macrophages and endothelial cells [61]. VEGF acts as one of the main inducers of angiogenesis by enhancing the proliferation and migration of endothelial cells and promoting sprouting from existent blood vessels [139]. HIF-1 signalling in hypoxic conditions also promotes the pro-inflammatory M1 phenotype of macrophages by inducing ATP production which in turn enhances cellular motility, invasiveness and bactericidal activity [140]. In hyperglycaemic conditions, angiogenesis is impaired due to the inhibition of HIF-1 signalling. Studies show that the expression of HIF-1α and VEGF in wounds of diabetic mice and HUVECs cultured in high glucose media is downregulated [29,141]. Considering the importance of HIF-1 signalling during wound healing and its inhibition during diabetic wound healing, makes HIF-1 signalling a promising target to promote the healing of diabetic ulcers.

### 7.3. Targeting HIF-1 Signalling to Promote Healing in Chronic Wounds

PHD2 and VHL are negative regulators of the HIF-1 signalling pathway. Therefore, targeting PHD2 and VHL is an appropriate strategy to enhance HIF-1 signalling in chronic wounds as they are characterised with low HIF-1 signalling activity. PHD2 and VHL inhibitors are being investigated for their potential in promoting diabetic wound healing by promoting the proliferation and migration of fibroblasts and inducing angiogenesis. Several treatment strategies are currently being investigated to inhibit the expression and activity of PHD2. Shaabani et al. showed that in vitro, transfection of fibroblasts cultured in hyperglycaemic media with the small interfering RNA (siRNA) siPHD2, reduced the expression of PHD2 and enhanced the expression of HIF-1α [22]. The increase in HIF-1α activity increased the proliferation and migration of fibroblasts.

A cell therapy approach is also currently being investigated for its potential in enhancing chronic wound healing. The application of MSCs with silenced PHD2 promoted wound healing in non-diabetic mice [142]. Further investigations are required to test the effect of siPHD2 in diabetic mice. Implantation of fibroblasts with silenced PHD2 expression in diabetic wounds of mice promoted wound healing by increasing angiogenesis as demonstrated by an increase in cluster of differentiation (CD) 31^+^ endothelial cells and micro-vessel density [30]. Dallas et al. applied an oligonucleotide-based approach to inhibit the expression of PHD2 in diabetic wounds of mice [31]. The delivery of the synthetic short hairpin RNA SG404 using layer-by-layer coated dressings to diabetic wounds of mice promoted HIF-1α expression. This in turn, stimulated neovascularisation and contributed to the healing of chronic wounds.

VHL inhibitors are less studied than PHD2 inhibitors. A recent study by Qiu et al. demonstrated that VH298 treatment promoted the proliferation and migration of fibroblasts and enhanced tube formation by HUVECs [23]. Furthermore, VH298 improved the rate of wound closure in streptozotocin-induced diabetic mice and increased the number of HIF-α and VEGF-A positive cells.

Several small molecules are being investigated for their effect in promoting angiogenesis in chronic diabetic wounds via HIF-1 signalling. Deferoxamine (DFO), 20(S)-protopanaxadiol, Iridium (III) complex Ia and arnebin (I) have been shown to increase the expression of HIF-1α and VEGF in wounds of diabetic mice, thus promoting angiogenesis and enhancing chronic wound healing [29,32,33,34]. In addition, HUVECs treated with asperosaponin VI also showed higher cell proliferation and migration, hence enhancing angiogenesis in vitro [143].

HIF prolyl-hydroxylase inhibitors (HIF-PHI) are a novel class of drugs and are currently being investigated for their effect on chronic wound healing [144]. FG-4592 also known as Roxadustat is a HIF-PHI, which acts by inhibiting the PHD domain, thus preventing the ubiquitinylation and subsequent degradation of HIF-1α [144]. Zhu et al. demonstrated that the treatment of diabetic wounds of mice with Roxadustat enhanced wound healing and promoted both HIF-1α and VEGF expression [35].

### 7.4. Targeting Novel Regulators of HIF-1 Signalling

Several long non-coding RNAs (lncRNAs), miRNAs and proteins were demonstrated to regulate HIF-1 signalling. The increase in expression of lncRNAs H19, GAS5 and Casc2 promoted wound healing in streptozotocin-induced diabetic mice and enhanced the expression of HIF-1α [141,145,146]. Similarly, miR-31-5p overexpression in wounds of diabetic mice showed the same effect [147]. In contrast, the downregulation of miR-210 and miR-217 resulted in enhanced wound closure in diabetic rats [31,148]. Recently, it was reported that Growth Differentiation Factor 11 (GDF11) and leucine-rich α-2 glycoprotein 1 (LRG1) are potential targets to enhance diabetic wound healing. Treatment of diabetic wounds with GDF11 promoted diabetic wound closure by enhancing angiogenesis as shown by the increase in expression of HIF-1α and VEGF [149]. Moreover, treatment of HaCaT cells with LRG1 enhanced their migration and promoted the stability of HIF-1α [150].

## 8. Conclusions

PI3K/AKT, Wnt/β-catenin, TGF-β, Nrf2, Notch and HIF-1 signalling are essential for proper wound healing. In chronic wounds these signalling pathways are dysregulated. Here, recent investigations with the aim of rectifying abnormal signalling and the promotion of healing in chronic wounds were highlighted. The treatments investigated included naturally derived, stem-cell-based, small molecules, oligonucleotide delivery, exosomes and peptide-based therapies (Figure 3). Moreover, several miRNAs, lncRNAs and proteins were discussed as potential targets to promote chronic wound healing. Although several types of treatments were investigated, their application in the clinic is still limited. Therefore, clinical trials should be carried out to further confirm their effectiveness. Furthermore, research on the role and regulation of signalling pathways during chronic wound conditions is essential to provide further opportunities for treatment development.

## Figures and Tables

**Figure 1 ijms-25-00050-f001:**
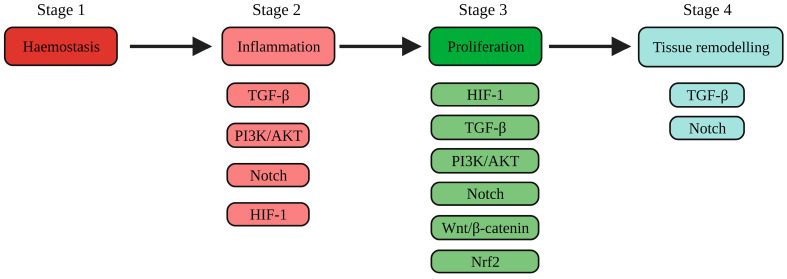
An overview of the wound healing process and the signalling processes involved in the inflammatory, proliferative, and remodelling phases. Abbreviations: PI3K: phosphoinostide-3-kinase; AKT: protein kinase B; Nrf2: nuclear factor erythroid 2-related factor 2; TGF-β: transforming growth factor-β; HIF-1: hypoxia inducible factor-1; Wnt: Wingless/integrated. Created with Biorender.com. Accessed on 5 December 2023.

**Figure 2 ijms-25-00050-f002:**
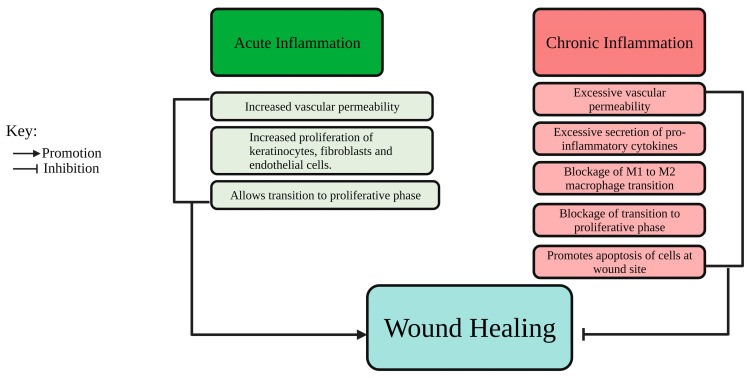
A comparison of acute inflammation in normal wound healing versus chronic inflammation exhibited by a chronic wound. Created with Biorender.com.

**Figure 3 ijms-25-00050-f003:**
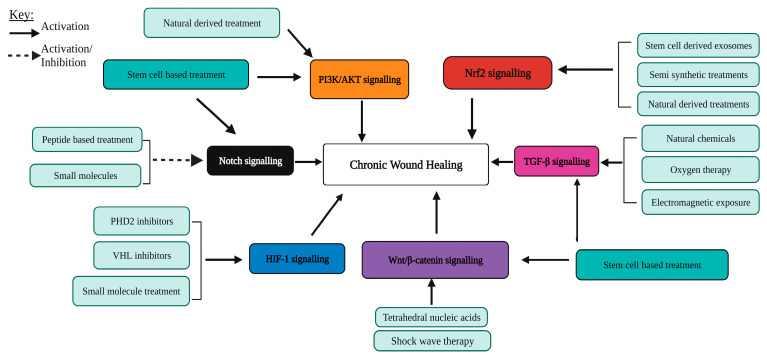
An overview of the effect of currently investigated treatment strategies in targeting PI3K/AKT, Nrf2, TGF-β, Notch and HIF-1 signalling pathways to promote chronic wound healing. Abbreviations: PI3K: phosphoinostide-3-kinase; AKT: protein kinase B; Nrf2: nuclear factor erythroid 2-related factor 2; TGF-β: transforming growth factor-β; HIF-1: hypoxia inducible factor-1; PHD2: prolyl hydroxylase domain protein 2; VHL: von Hippel–Lindau. Created with BioRender.com. Accessed on 5 December 2023.

**Table 1 ijms-25-00050-t001:** The activation of signalling pathways required to enhance chronic wound healing.

Processes Required to Enhance Chronic Wound Healing	Activation of Pathways Involved in the Processes of Wound Healing
The promotion of M1 to M2 macrophage transition	TGF-β [13] and Notch [14]
The promotion of keratinocyte proliferation	PI3K/AKT [15]
The promotion of fibroblast proliferation and migration	PI3K/AKT [15,16,17], Wnt/β-catenin [18], TGF-β [19,20], Nrf2 [21] and HIF-1 [22,23]
The promotion of angiogenesis	PI3K/AKT [15,24,25], Wnt/β-catenin [26,27], Notch [28] and HIF-1 [23,29,30,31,32,33,34,35]
The promotion of remodelling	TGF-β [36,37,38,39,40,41,42]
The promotion of keratinocyte migration	TGF-β [43]

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
