# Peer review of "Targeting Signalling Pathways in Chronic Wound Healing"

_ijms, 2023, doi:10.3390/ijms25010050_

Round 1
Reviewer 1 Report
Comments and Suggestions for Authors
Regretfully, I must write a negative review on this paper.
First, a main goal of a review is to summarize the facts already known by putting them together in a specific context, and this could actually increase the knowledge we have on a topic. However, sections of this review are organized as a plain list of results. For example, in section 2.2 I see just a list of statements from the paper abstracts, connected with words like “furthermore” and “similarly” : Xiao et al. reported that <…>. Furthermore, <…>. Similarly, <…>. In a study by Huang et al., <…>. But I failed to find any conclusion out of all these furthermore’s, and from my point of view the current version of the paper does not go beyond a compilation. The thought is still, it does not develop anywhere.
The only “kernel” which I noticed in the paper is a continuous mentioning of the “natural components” and their role in the wound healing. I would not state them all here, just a brief list: section 2.2 (starting from line 147) in completely devoted to “natural extracts”, “ruyi jinhuang powder” is mentioned in line 200, maggot therapy in line 274, Chinese Herbal medicine Huiyang shengji in line 279, extract from indigo leaves in line 286, “other naturally derived treatments” (line 296), and even something as exotic as “the human placental extract (HPE)” (line 329). But even this line is not proceeded any further from simple list of papers where this and that was shown.
Second, figures in the manuscript look much more like slides from presentation, and they actually contain limited information. I would expect much more details and more conceptualization from figures in a review.
Author Response
Dear Reviewer,
Thank you for your constructive feedback. The images were improved using BioRender instead of powerpoint. The main aim of this review was to provide an update on the approch of targeting dysregulated pathways to promote chronic wound healing. However, we have taken into account your feedback and hence we discussed the limitations of this area in research together with further changes highlighted in yellow.
Kind regards.
Reviewer 2 Report
Comments and Suggestions for Authors
It is well known that chronic wounds are severely affecting the life quality of patients, which is already an economic burden to the healthcare system across the world with huge treatment cost on this type of wounds. From the perspective of individual, patients also pay a large spending for related treatments. Currently, such wounds are commonly treated using a combination of different strategies. Although there are several treatment options under investigation, few have been approved by the Food and Drug Administration. Therefore, a review summarizing current understanding to chronic wound healing is essential to drug discovery and clinical treatment. Based on this background, this manuscript detailedly reviewed several specific signaling pathways that are involved in chronic wound healing, making this manuscript distinct from other reviews.
Authors summarized current research in this field through citing newest research references. They firstly summarized the physiology of wound healing as well as Pathophysiology of wound healing with detailed introduction from citing many references. They secondly gave discussions of several specific singling pathways involved in the processes of chronic wound healing, including The canonical Wnt/β-catenin pathway, TGF-β signalling, Nrf2 signalling pathway, Notch signalling pathway, HIF-1 signalling pathway. For each of these signaling pathways, they cited related research papers to provide evidence to review current progresses in this field. In this manuscript, authors totally drew two figures to help understand the theme of this manuscript. Detailed mechanism summary about these signaling pathways advance further our understanding in the field of chronic wound healing. This review manuscript nearly included all recently related research in this field. It can represent a newest review for this field.
Based on evidence authors provided in this review, I think authors show a good background to understand these signaling pathways in the chronic wound healing. This review can be a good reference for readership who is interested in mechanism of chronic wound healing. More importantly, authors also discussed how current research progress in chronic wound healing can be used for the development of therapeutic strategies, which can benefit the healthcare system worldwide. Overall, I think this manuscript was organized and written well. The logic of this manuscript is also reasonable from my review. Evidence is sufficient to support authors to make their conclusions. The English quality is also good without major edition. Only several grammars in the text should be noticed, in which I will give my comments in the following. So, after reading through this manuscript carefully, I only have several minor comments that can be found as below. They are mainly related to correction of words. I hope these minor comments can help improve the manuscript.
Minor comments:
1. line 76, change “promote” to “promotes”
2. line 116, change “produce” to “produces”
3. line 154, change “resulted” to “resulting”
4. line 199, change “effective” to “being effective”
5. line 306, how to understand “positively regulating the TGF/Smad signaling pathway”? Do authors have better words to replace “positively”?
6. line 313, just to make sure whether the word “cap’n collar” is correct or not.
7. line 333, change “demonstrates” to “demonstrated”
8. line 555, gramma of the sentence “Consequently, promoting neovascularisation and 555 chronic wound healing” needs correction.
Comments on the Quality of English Language
English is good.
Author Response
Dear Reviewer,
Thank you very much for your constructive feedback. We have taken into account your suggestions and changes were highlighted in yellow.
Reviewer 3 Report
Comments and Suggestions for Authors
The topic is very interesting in order to understand the intrinsic mechanisms involved in wound healing.
The work is well organised and written in an accurate and precise way.
From a scientific point of view the work is sound and not misleading.
The references are appropriate; however, I suggest the inclusion of the following citations:
Tetè G, D'Orto B, Nagni M, Agostinacchio M, Polizzi E, Agliardi E. Role of induced pluripotent stem cells (IPSCS) in bone tissue regeneration in dentistry: a narrative review. J Biol Regul Homeost Agents. 2020 Nov-Dec;34(6 Suppl. 3):1-10.
Capparè P, Tetè G, Sberna MT, Panina-Bordignon P. The Emerging Role of Stem Cells in Regenerative Dentistry. Curr Gene Ther. 2020;20(4):259-268. doi: 10.2174/1566523220999200818115803.
English language fine. No issues detected
Author Response
Dear Reviewer,
Thank you very much for your opinions and interest in our paper. Unfortunately I was not able to retrieve the two suggested articles and hence we were not able to include them.
Kind regards.